# The Cancer Genome: Paradigm or Paradox?

**DOI:** 10.3390/cancers13040674

**Published:** 2021-02-08

**Authors:** Shi-Ming Tu

**Affiliations:** Department of Genitourinary Medical Oncology, Unit 1374, The University of Texas MD Anderson Cancer Center, 1155 Pressler Street, Houston, TX 77030-3721, USA; stu@mdanderson.org; Tel.: +1-(713)-563-7268; Fax: +1-(713)-745-1625

**Keywords:** cancer genome, driver mutation, precision medicine, targeted therapy, cancer stem cells

## Abstract

**Simple Summary:**

The observation that genetic mutations often do not cause cancer or disease in the phenomena of mosaicism, clonal hematopoiesis of indeterminate potential (CHIP), and heteroplasmy provides us with important clues about the origin and nature of cancer. We should be wary that the cancer genome may lead us astray to the wrong destination on a bad expedition unless we adopt the right cancer theory to elucidate it, and adhere to the proper scientific method to investigate it.

**Abstract:**

Nowadays, many professionals are sequencing the DNA and studying the cancer genome. However, if the genetic theory of cancer is flawed, our faith in the cancer genome will falter. If gene sequencing is only a tool, we should question what we are making or creating with this tool. When we do not have the right cancer theory at our disposal, we cannot be sure that what we create from the cancer genome is meaningful or useful. In this article, we illustrate that mosaicism, CHIP, and heteroplasmy dispute our traditional perspectives about a genetic origin of cancer and challenge our current narratives about the cancer genome. We caution that when we have the wrong cancer theory, big data can provide poor evidence. Precision medicine may become rather imprecise. Targeted therapy either does not work or work for the wrong reasons. The cancer genome thus becomes a paradox rather than a paradigm.

Nowadays, we see splendor and hear glory with regard to the cancer genome. The sequencing of DNA and search for driver genetic mutations is widespread. Precision medicine is taught and targeted therapy is practiced. According to the genetic theory of cancer, we should be finding a panacea for cancer and reaching the nirvana of cancer care. However, if the genetic theory of cancer is flawed, our faith in the cancer genome will falter.

A recent paper in the *New England Journal of Medicine* highlights the dilemma with regard to DNA in breast cancer risk [1]. Dorling et al. detected pathogenic genetic variants in 5.01% of breast cancer cases (*n* = 32,247) and 1.63% of the control group (*n* = 32,544). Of the 28 putative breast cancer genes investigated, only 12 showed clear evidence of an associated cancer risk. The meaning and the implications of the cancer genome are unclear when the majority of alleged cancer genes are not associated with cancer risk and when a significant number of people without cancer actually harbor cancer genes. Although we still have faith in the value of the cancer genome for the purposes of cancer diagnosis, prognosis, and therapy, perhaps we should temper that faith by examining it in the right scientific mindset, e.g., in the proper cellular context and with a pertinent cancer theory.

In this article, we ask a simple question, pose an elemental problem, and offer a modest proposal about the cancer genome. We demonstrate that without the blessing of a correct theory about the origin or nature of cancer, driver mutations may represent a fantasy or fallacy, precision medicine and targeted therapy may become a bust rather than a boom, and the cancer genome is a paradox rather than a paradigm. We illustrate that mosaicism, CHIP, and heteroplasmy dispute our traditional perspectives about a genetic origin of cancer and challenge our current narratives about the cancer genome. We should be wary that the cancer genome may lead us astray to the wrong destination on a bad expedition unless we adopt the right cancer theory to elucidate it and adhere to the proper scientific method to investigate it [2,3,4].

## 1. Fantasy or Fallacy

In cancer research, the genome has become gospel. In cancer therapy, precision medicine is the Holy Grail.

In 1914, Boveri first linked cancer to genetic aberrations [5]. In 1988, Vogelstein advanced the genetic theory of cancer in his model of multistep carcinogenesis [6]. In 2001, Landers et al. and Venter et al. sequenced the human genome [7,8].

There is great expectation that the cancer genome will improve cancer diagnosis, elevate cancer screening, and ameliorate cancer therapy.

There has been a bold prediction that the cancer genome will facilitate precision medicine, expedite targeted therapy, and optimize personalized care.

Although promising treatment options have been produced and definitive clinical benefits have been provided, what we have reaped from the cancer genome does not seem conmmensurate with what we have sown. Most intractable cancers still recur, and most lethal cancers still kill. Perhaps our infatuation with the cancer genome and our obsession with the genetic theory of cancer is at the root of a plethora of modest gains and marginal returns in cancer care.

It is time for us to reexamine the records and revisit the evidence about the cancer genome and precision medicine. Perhaps we have misunderstood the whole bible and misinterpreted the entire scripture with regard to the basic origin and nature of cancer.

## 2. Simple Question

When we treat cancer patients everyday on a practical level in the clinic, we can make interesting observations and ask obvious questions.

Although more and more clinically certified gene sequencing assays are becoming available for patient care, most of us do not know the inner workings and fine details of their applications or implications. We may not realize that when tumor specimens are sequenced, the DNA from both cancer and non-tumor cells is being analyzed. We may not appreciate that when tumor specimens are biopsied, the samples obtained at different sites in the same tumor may comprise distinct cellular components and harbor disparate genetic profiles.

It is evident that the results from a preponderance of the genetic tests may be of questionable value and utility. Currently, most genetic mutations are not actionable. When they are, the clinical benefits from targeted therapy tend to be minor, minimal, and momentary [9].

Furthermore, when we screen patients and check genetic mutations for diagnostic and therapeutic purposes, we may not be able to duplicate the results using different assays, or even the same assay [10,11]. When we happen to perform serial tumor or liquid biopsies, we may see variable results over time [12].

In many respects, the evidence is confounding. The numbers simply do not add up. Complete or even common sense may not always be obtained from big data.

Although we preach precision medicine, it is clear that cancer is not very precise. We have learned that cancer is dynamic, erratic, and byzantine. In fact, it is precisely the opposite of precise.

Although we practice targeted therapy, we may be obligated to treat many targets and combine targeted therapies. In fact, we are leaning towards adding chemotherapy to the list of targeted therapies, the antithesis of and an anathema to the very idea of targeted therapy.

Like Wolfgang Pauli famously said, “This is not only not right, it is not even wrong.”

That is the problem.

## 3. Elemental Problem

Perhaps the problem is more than elemental, because we have been embracing a poor theory and endorsing incorrect ideas rather than just promoting bad platforms and performing the wrong assays.

After all, on studying cancer at a scientific level in the laboratory, we have learned that maintenance of oncogene expression is not a prerequisite for many malignant phenotypes [13]. Most genetic mutations occur at a low frequency in many cancer types [14]. Only about 30% of mutations are shared among multiple sites in the primary tumor and metastatic lesions [15]. Up to eight biopsies are required to capture a majority (and not even all) of the putative driver mutations [16].

It is evident that lethal subclones can be dormant or suppressed, and may not be detectable from the outset [17,18]. When a lethal subclone happens to be present at the outset, its dominance may wax and wane [19]. Importantly, it may very well be an overall tumor makeup rather than the individual subclones that determine the natural history and clinical course of a particular tumor [20].

Therefore, when we consider the significance and relevance of the cancer genome during carcinogenesis, the cellular context is pivotal. Progenitor stem-like cells, as compared with progeny differentiated cells, have disparate mitigating (e.g., DNA repair, antioxidants) and aggravating (e.g., reactive oxygen species, inflammatory factors) properties and processes that affect formation of genetic mutations [21,22]. Importantly, the same genetic mutations may elicit disparate effects and engender disparate entities in a progenitor stem-like cell compared with a progeny differentiated cell.

Perhaps many cancer researchers and clinicians have ignored or neglected a basic fundamental oncological principle: that cancer is above and beyond the genome.

Unfortunately, this is a major systemic issue rather than a mere academic problem.

## 4. Boom or Bust

It seems ironic that a scientific boom may turn out to be a clinical bust, but a scientific bust can still provide a financial boom (albeit temporary).

It is true that scientific ingenuity and innovations create untold opportunities that are good for the economy and for our society as a whole. Even bad science is good for the career of many scientists and clinicians, and for the pocketbooks of many drug and tech companies. It may also be good for a minority of patients (perhaps for the wrong reasons). However, this still does great injustice and brings no solace to a majority of patients who do not benefit and who may be harmed by our scientific misadventures and clinical miscues.

It is true that science, like finance, will correct itself. Scientists and clinicians receive their share of fame, while drug and tech companies have their fill of fortune. However, for the patients who trust our scientific endeavors, any misadventures and miscues on our part do not appear fair nor trivial at a personal level. This is because to them, it is a matter of life or death. There may be an endless passage of physical misery, psychological agony, and financial insolvency.

It is also true that by force of our nature and culture, we have become enamored of vogue, ensnared by norm, and enslaved by lore. Unknowingly, we submit to peer pressure and herd mentality. Unwittingly, we espouse the genetic theory of cancer, empower genetic testings of cancer, enforce precision medicine, and enable targeted therapy.

## 5. Mosaicism

DNA mosaicism exposes a potential flaw in the idea of a cancer genome. Although DNA mutations cause a mosaic, they are actually a common and normal occurrence in the natural world and in healthy people.

In medieval Europe, forest travelers noticed odd growths sprouting from tree trunks. The Germans called these growths “Hexenbesen”—witches’ broom. In 1904, botanists in Northern Canada stumbled on an outgrowth of a white spruce [23], a species that can grow to 10 stories high. Today, dwarf Alberta spruce is a landscaping favorite, growing up to 10 feet tall. In 1906, a farmer in Florida discovered that fruits from an odd branch on a grapefruit tree had pink rather than white flesh [24]. Seeds from those fruits have produced pink grapefruit ever since.

It seems that not only is there an abundance of mosaics in nature, they are prevalent in us, too. Not only does the genome vary from person to person, but DNA may also vary from cell to cell in the same person. We have been taught and many of us still believe that all cells in our body carry and share an identical set of genes from birth and for the rest of our lives. The phenomenon of mosaicism reminds us that those ideas may be obsolete.

Therefore, even though cancer involves mutations, these mutations do not always or necessarily cause cancer. Mosaicism suggests that cancer is a problem above and beyond genetic mutations. It alludes to a recurring theme and overriding observation that the cell in which the mutations reside and from which the cancer originates also matters and is paramount.

## 6. CHIP

CHIP also suggests that something is amiss with the idea of a cancer genome.

Khiabanian et al. noticed that in some solid tumor specimens, there were mutations in genes commonly found in myeloid disorders [25]. These mutations were often present at low levels (i.e., in low abundance) compared with the amount of tumor. They hypothesized that these mutations were due to a separate age-related mechanism rather than a tumor-related condition, in which mutations were found in “normal” blood cells, but no hematological malignancy was ever (or yet) diagnosed.

They analyzed 113,079 solid tumors to identify the genes with the highest rate of detected mutations. They discovered that only four mutations in 257 genes were detected at higher rates in older patients’ solid tumors, independent of their cancer type. Importantly, these four genes (*DNMT3A, TET2, SF3B1, and ASXL1*) were known to be associated with CHIP.

Subsequently, they examined patient-matched peripheral blood samples and demonstrated that mutations in the *DNMT3A and TET2* genes had a blood-related rather than tumoral origin in 11 (79%) of 14 cases.

In particular, there was a patient diagnosed with two independent lung cancers of different histological findings and distinct sets of genomic alterations. The only commonality between the specimens involved two identical mutations in the *DNMT3A and TET2* genes, which were detected at low levels. When they examined the blood cells from this patient, they confirmed that the same mutations were present in the blood cells rather than in the lung cancer cells.

## 7. Heteroplasmy

In heteroplasmy, in which mitochondrial DNA (mtDNA) variants coexist in a single cell or among cells within an individual, we witness another revealing discrepancy in the idea of a cancer genome.

There are only two copies of nuclear DNAs, but hundreds to thousands copies of mtDNA in each single human cell. These mtDNAs may differ from one other due to inherited or somatic mutations.

If the DNA is a blueprint or the master plan of our whole being, then it is vital for us to have many ways to ensure its accuracy, reliability, and durability.

One way to keep the DNA intact is to be equipped with robust mechanisms for repair. The other way to keep the DNA functional is to have copious reserves and redundancies.

Hence, not only do we have the same gene on each of the two paired chromosomes (except for the sex chromosomes), but we may have multiple genes on the same chromosome, and similar genes on different chromosomes. This way, when a particular gene becomes defective, others can make up or may take over its vital functions.

This idea of DNA reserves and redundancies means that certain genetic mutations may not always be detrimental. This accounts for the observation of countless incidental or passenger genetic mutations that appear to be innocuous, if not irrelevant.

Paradoxically, pathogenic heteroplasmy is pervasive in healthy individuals [26]. About 90% of individuals carry at least one heteroplasmy. At least 20% harbor heteroplasmies implicated in disease.

However, most of these pathogenic mutations appear to be harmless, because they are either completely neutralized or well compensated by the presence of wild type mtDNA.

It seems that heteroplasmy is another example of an ambiguity in the idea of a cancer genome and fallibility in the theory of a genetic origin of cancer.

## 8. Genetic Mutations Cause Cancer, Except Many Do Not

Because cancer involves mutations, one assumes that mutations cause cancer. Because genetic mutations cause mosaicism, and mosaicism is associated with certain disease, one assumes that genetic mutations cause disease. However, when mosaics are also common occurrences in healthy people, and when genetic mutations often do not cause cancer or disease, it seems that what one assumes is incorrect and the idea should be corrected.

Perhaps one can say that a specific mutation may be the fuse if not the explosion that ignites the whole process of carcinogenesis. One can also say that unknown or additional mutations may have initiated (if not instigated) cancer. Perhaps our epistemic closure or confirmation bias about a genetic origin of cancer is incorrigible.

Stratton et al. reported that every person acquires early embryonic mutations, each of which is present in a sizable proportion of one’s cells [27]. They hypothesized that about three of these mutations arise each time a cell divides when the person is still an embryo, and that embryonic cells pass the mosaic genetic signatures down to descendant cells in various tissues and organs. Despite the innumerable mutations in their genome from the very beginning of their lives, the 241 subjects in the study apparently did not have cancer over a prolonged period and the mutations probably would not have caused cancer over the life course.

Walsh et al. also demonstrated mosaics in the brains of healthy people. In one study, they examined individual neuron cells from the brain of a 17-year-old boy who had died in a car accident [28]. They sequenced the DNA in each neuron and compared it with DNA in cells from the boy’s liver, heart, and lungs. They found that each neuron had hundreds of mutations not present in cells from other organs, and many mutations were not shared by other neurons in the same organ. Using the mutations as markers, they traced the genetic genealogy and mosaic heritage of each cell. Strangely, they found cells in the heart with the same mutational signature as that in some brain neurons. It appears that five lineages of cells, each with a distinct set of mutations, had formed when he was an embryo. Cells from different lineages migrated in different directions and merged to form different organs. In essence, each organ comprises clusters of mosaic cells of related but distinct cellular origins and destinies.

In principle, mutations introduce randomness and mosaics produce diversity in different people: “The same zygote would never develop exactly the same way twice.”

However, most mutations do not and will not cause cancer.

Why not? That is the critical observation and a key question.

## 9. Chimpanzees and Twins

A critical observation about the intricacies of the cancer genome becomes apparent when we discover substantial similarities in the genetic makeup between chimpanzees and humans and when we detect subtle differences in the DNA profiles of identical twins.

Hence, with an only 1% difference in the protein-coding genome between the two species [29], a human may be similar enough to a chimpanzee when it concerns their genotypes, but they are certainly very dissimilar when we consider their phenotypes (Figure 1). Conversely, even in identical twins DNA may not be identical. On average, monozygotic twins differ by about five early developmental mutations [30]. From those mutations, one could trace the origin of a twin from a single cell or several cell lineages (not unlike what Stratton et al. and Walsh et al. have shown).

A key question about the implications of the cancer genome is whether we have framed a pertinent hypothesis and designed proper experiments to test the hypothesis according to the scientific method.

When we formulate a flawed scientific theory and generate hypotheses from experiments rather than test hypotheses with experiments, the scientific evidence may seem flawless and formidable, but it is also self-fulfilling and self-serving.

For example, how we select the controls in an experiment depends on the hypothesis we propose and the question we ask. When we hypothesize that genetic mutations cause a particular cancer, we start with the cancer and look for mutations in that cancer. However, when we hypothesize that the cellular context (i.e., origin in a progenitor stem-like cell or progeny differentiated cell) of those genetic mutations is paramount, we start with the mutations and determine whether they actually cause cancer (as done by Dorling et al.).

In many respects, the studies on chimpanzees and twins defy conventional thinking and challenge cherished beliefs. They invite an alternative perspective that cancer is a genetic disease. They incite a contrasting narrative that the cancer genome could be a misconception, if not a misnomer.

## 10. A Modest Proposal

When we adopt the genetic theory of cancer, we believe that defective *p53* and *BRAF* are responsible for the formation of melanoma, and *p53* and *RB1* loss are involved in the development of small cell carcinoma. However, when we support a stem-cell theory of cancer, we perceive the same genetic defects in a different light and prescribe them a different role [31,32].

After all, a cell with defective *p53* and *BRAF* does not become a melanoma unless the cell also happens to be a crestin-expressing stem cell [33]. Small-cell carcinoma with *p53* and *RB1* loss reverses its phenotype when it stops expressing its stem-ness gene, *SOX2* [34].

Suppose a progenitor stem cell that transforms into a malignant melanoma or small-cell carcinoma has always had inactive tumor suppressor genes (such as *p53*). In this case, what is the significance of a TP53-null or -negative stem cell versus a malignant cell?

In other words, if malignant cells are derived from stem cells, and both cells are intrinsically TP53-null or -negative, then TP53 loss makes all the difference in the genetic theory of cancer, but does not really matter in the stem-cell theory of cancer.

When we advocate the genetic theory of cancer, defective *p53* and *BRAF* cause melanoma, and *p53* and *RB1* loss cause small-cell carcinoma. However, when we advance a stem-cell theory of cancer, aberrant progenitor stem cells with certain genetic defects become malignant, but aberrant progeny non-stem cells with the same or similar genetic defects do not.

We predict that a benign mole in the skin with defective *p53* and *BRAF* will never become a melanoma because it does not originate from a stem-like cell. Similarly, benign prostatic hyperplasia with *p53* and *RB1* loss will not convert into small cell carcinoma of the prostate, for the same reason.

## 11. Conclusions

Nowadays, gene sequencing on tumors of all stripes and colors is common. However, if gene sequencing is only a tool, there is an issue as to exactly what we are making or creating with this tool. When we do not have a good cancer theory, there is a good chance that what we infer from the cancer genome may be meaningless or worthless.

Unfortunately, there is reticence within the scientific community when our scientific research does not adhere to the scientific method, and denial within the clinical arena when our seminal basic research does not translate into effective treatments.

The observation that genetic mutations often do not cause cancer or disease in the phenomena of mosaicism, CHIP, and heteroplasmy provides us important clues about the origin and nature of cancer.

According to the genetic theory of cancer, precision medicine and targeted therapy should be precise and relatively simple. For a few malignancies, such as chronic myelogenous leukemia, this is indeed true. However, for the vast majority of cancers, any treatment benefit based on or derived from this theory remains incremental rather than exponential, and elusive rather than curative.

According to a stem-cell theory of cancer, integrated medicine and multimodal therapy could be transcendental and transformative. When we treat the genetics and epigenetics of cancer, the progenitor stem-like and progeny differentiated cancer compartments, and the ubiquitous tumor microenvironment, we may improve the clinical outcome of established modalities and enhance the clinical efficacy of approved therapies.

We caution that when we have the wrong cancer theory, big data may provide bad evidence. Precision medicine becomes rather imprecise. Targeted therapy may either not work or work for the wrong reasons. The cancer genome is a paradox, rather than a paradigm.

## Figures and Tables

**Figure 1 cancers-13-00674-f001:**
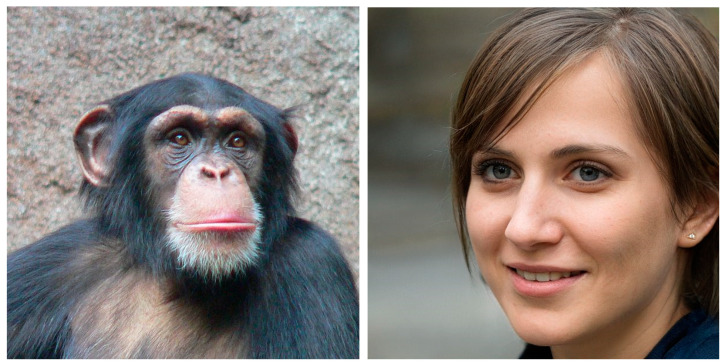
Similar genotype, dissimilar phenotype. A chimpanzee (photo by Thomas Lersch, 2005) and human (the person in this photo does not exist, but has been generated by artificial intelligence (StyleGAN) based on an analysis of portraits). https://commons.wikimedia.org/wiki/File:Chimpanzee-Head.jpg (accessed on 1 January 2021). https://commons.wikimedia.org/wiki/File:Woman_7.jpg (accessed on 1 January 2021).

## Data Availability

This Perspective article does not report any new data.

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
