# Peer review of "The Cancer Genome: Paradigm or Paradox?"

_cancers, 2021, doi:10.3390/cancers13040674_

Round 1
Reviewer 1 Report
In this interesting paper, the Author performs an assessment of the progress made in understanding of carcinogenesis and the efficacy of cancer treatment. Although the tremendous achievements in this area have obviously been pointed out by the Author, he convincingly delineates all the failures of single-sided approaches, either based on genetic theory of cancer, or the role of single mutations of critical genes, or even the targeted controlled delivery of therapeutic drugs. All the failure notices and inadequacies discussed are well documented and the criticism of simplistic treatments of challenging problems, we are facing with this deadly disease, supported by literature references. The assessment of the approaches to cancer therapy presented in this paper is very valuable for general Readership, as it enables gaining a better understanding of the challenges involved. Also, it motivates scientists and clinicians to boost their efforts to solve the problems in their complexity, rather than holding to simplified models of the disease. This beautifully written PERSPECTIVE end with a general conclusion, stating that the “integrated medicine and multimodal therapy” should enhance our chances for curative treatments. I recommend the paper for publication after minor revisions addressing the comments presented below:
1. The first problem which needs to be expanded is the problem of mutations. There is an impression from the paper that a single or dual mutation is the only mode of genomic changes and, in the simplistic view, this type of change, if it concerns p53 or other critical genes, induces carcinogenesis. In opposite view, taking into account repair mechanisms, ample number of gene copies and redundancy, these mutations do not matter much and may never induce the carcinogenesis. However, it is known that long term mild inflammations, often overlooked, may cause massive attacks by reactive oxygen species (ROS), released by the immune system, inflicting a severe damage to DNA, causing large number of mutations, and lead to cancer. Therefore, in contrast to single mutation-causing carcinogenesis in genetic cancer theory, we should also consider massive mutations due to organism’s own defense ROS system. With the massive mutations, the probability that a stem cell in the tissue acquires certain genetic defects is high and that cell may become malignant. Therefore, a cancer is formed when massive inflammation/mutations occur, but not when a single random mutation, such as the one considered in genetic cancer theory, happens. Now, ROS is generated right in the cells, in mitochondria (MTs). By controlling the respiration processes (the main source of ROS) and avoiding hypoxia, it becomes possible to prevent mutations of the nucleic acids. On the other hand, by enhancing ROS generation, an apoptosis of the cancer cell can be induced. Recently, the investigations of breathing processes in MTs and the associated large volume expansion/contraction of MTs, have been performed by studying the ion dynamics in MTs immobilized on the surface of a piezosensor. The technique proposed in these investigations enables to examine the effects of drugs on breathing processes in MTs and hypoxia by analyzing piezometric responses of the sensor (Biosensors and Bioelectronics, 2017, 88: 114-121). This relevant literature reference, should be cited.
2. The protective effects of glutathione and other antioxidants against pre-mutagenic DNA damage by strong oxidants or HO* radical generating species (peroxides, Fenton reaction, etc.) serve also as an example that both the environmental and exogenic compounds can intervene and protect nucleic acids against damage to vulnerable genes leading to mutations (Mutation Research - Fundamental and Molecular Mechanisms of Mutagenesis; 2012, 735: 1– 11).
3. There are no typographical or English errors.
Author Response
In this interesting paper, the Author performs an assessment of the progress made in understanding of carcinogenesis and the efficacy of cancer treatment. Although the tremendous achievements in this area have obviously been pointed out by the Author, he convincingly delineates all the failures of single-sided approaches, either based on genetic theory of cancer, or the role of single mutations of critical genes, or even the targeted controlled delivery of therapeutic drugs. All the failure notices and inadequacies discussed are well documented and the criticism of simplistic treatments of challenging problems, we are facing with this deadly disease, supported by literature references. The assessment of the approaches to cancer therapy presented in this paper is very valuable for general Readership, as it enables gaining a better understanding of the challenges involved. Also, it motivates scientists and clinicians to boost their efforts to solve the problems in their complexity, rather than holding to simplified models of the disease. This beautifully written PERSPECTIVE end with a general conclusion, stating that the “integrated medicine and multimodal therapy” should enhance our chances for curative treatments. I recommend the paper for publication after minor revisions addressing the comments presented below:
Thank you for your insightful and positive comments.
- The first problem which needs to be expanded is the problem of mutations. There is an impression from the paper that a single or dual mutation is the only mode of genomic changes and, in the simplistic view, this type of change, if it concerns p53 or other critical genes, induces carcinogenesis. In opposite view, taking into account repair mechanisms, ample number of gene copies and redundancy, these mutations do not matter much and may never induce the carcinogenesis. However, it is known that long term mild inflammations, often overlooked, may cause massive attacks by reactive oxygen species (ROS), released by the immune system, inflicting a severe damage to DNA, causing large number of mutations, and lead to cancer. Therefore, in contrast to single mutation-causing carcinogenesis in genetic cancer theory, we should also consider massive mutations due to organism’s own defense ROS system. With the massive mutations, the probability that a stem cell in the tissue acquires certain genetic defects is high and that cell may become malignant. Therefore, a cancer is formed when massive inflammation/mutations occur, but not when a single random mutation, such as the one considered in genetic cancer theory, happens. Now, ROS is generated right in the cells, in mitochondria (MTs). By controlling the respiration processes (the main source of ROS) and avoiding hypoxia, it becomes possible to prevent mutations of the nucleic acids. On the other hand, by enhancing ROS generation, an apoptosis of the cancer cell can be induced. Recently, the investigations of breathing processes in MTs and the associated large volume expansion/contraction of MTs, have been performed by studying the ion dynamics in MTs immobilized on the surface of a piezosensor. The technique proposed in these investigations enables to examine the effects of drugs on breathing processes in MTs and hypoxia by analyzing piezometric responses of the sensor (Biosensors and Bioelectronics, 2017, 88: 114-121). This relevant literature reference, should be cited.
- The protective effects of glutathione and other antioxidants against pre-mutagenic DNA damage by strong oxidants or HO* radical generating species (peroxides, Fenton reaction, etc.) serve also as an example that both the environmental and exogenic compounds can intervene and protect nucleic acids against damage to vulnerable genes leading to mutations (Mutation Research - Fundamental and Molecular Mechanisms of Mutagenesis; 2012, 735: 1– 11).
Agree with reviewer that disparate mitigating (e.g., DNA repair, antioxidants) and aggravating (e.g., reactive oxygen species, inflammatory factors) properties and processes22,23 would affect formation of genetic mutations.
We addressed this problem of mutations by adding the recommended references and putting it in a cellular context under the section of “Elemental problem”: the same genetic mutations may elicit disparate effects and engender disparate entities in a progenitor stem-like cell compared with a progeny differentiated cell.
- There are no typographical or English errors.
Reviewer 2 Report
The author has outlined the reasons to suggest why cancer genomics is more a paradox rather than a Paradigm. The ideas have been very coherently put forth with caution to scientists and clinicians on how the concept of genomics needs to be maneuvered in cancer precision medicine. The ideas are boldly stated but in a convincing manner.
The author could say in a few sentences the limits that could be set for using cancer genomics in precision medicine that would help prevent it from becoming a paradox.
Author Response
The author has outlined the reasons to suggest why cancer genomics is more a paradox rather than a Paradigm. The ideas have been very coherently put forth with caution to scientists and clinicians on how the concept of genomics needs to be maneuvered in cancer precision medicine. The ideas are boldly stated but in a convincing manner.
Thank you for your succinct and positive comments.
The author could say in a few sentences the limits that could be set for using cancer genomics in precision medicine that would help prevent it from becoming a paradox.
Thank you for this great suggestion, because it highlights the dilemma and may help us solve a paradox of the cancer genome.
We introduced “A recent paper in the New England Journal of Medicine highlights this dilemma about the DNA in breast cancer risk.1
…What is the meaning and what are the implications of the cancer genome when a majority of those alleged cancer genes are not associated with cancer risk and when a significant number of people without cancer actually harbor cancer genes.”
Perhaps one way to prevent the cancer genome from becoming a paradox is by “examing it under the right scientific mindset, e.g., in the proper cellular context and with a pertinent cancer theory.”
We moved the chimpanzee and twin data to a new section before “A modest proposal” in which we emphasized cellular context and the scientific method: …how we select the controls in an experiment depend on the hypothesis we pose and the question we ask. When we hypothesize that genetic mutations cause a particular cancer, we start with the cancer and look for mutations in that cancer. However, when we hypothesize that cellular context (i.e., origin in a progenitor stem-like cell or progeny differentiated cell) of those genetic mutations is paramount, we start with the mutations and determine whether they actually cause cancer.